# Genetic Strain Diversity of Multi-Host RNA Viruses that Infect a Wide Range of Pollinators and Associates is Shaped by Geographic Origins

**DOI:** 10.3390/v12030358

**Published:** 2020-03-24

**Authors:** Jana Dobelmann, Antoine Felden, Philip J. Lester

**Affiliations:** School of Biological Sciences, Victoria University of Wellington, Wellington 6012, New Zealand; antoine.felden@vuw.ac.nz (A.F.); phil.lester@vuw.ac.nz (P.J.L.)

**Keywords:** honey bee virus, pollinator, bee associate, *Apis mellifera*, *Linepithema humile*, DWV, KBV, Moku virus, invasive species

## Abstract

Emerging viruses have caused concerns about pollinator population declines, as multi-host RNA viruses may pose a health threat to pollinators and associated arthropods. In order to understand the ecology and impact these viruses have, we studied their host range and determined to what extent host and spatial variation affect strain diversity. Firstly, we used RT-PCR to screen pollinators and associates, including honey bees (*Apis mellifera*) and invasive Argentine ants (*Linepithema humile*), for virus presence and replication. We tested for the black queen cell virus (BQCV), deformed wing virus (DWV), and Kashmir bee virus (KBV) that were initially detected in bees, and the two recently discovered *Linepithema humile* bunya-like virus 1 (LhuBLV1) and Moku virus (MKV). DWV, KBV, and MKV were detected and replicated in a wide range of hosts and commonly co-infected hymenopterans. Secondly, we placed KBV and DWV in a global phylogeny with sequences from various countries and hosts to determine the association of geographic origin and host with shared ancestry. Both phylogenies showed strong geographic rather than host-specific clustering, suggesting frequent inter-species virus transmission. Transmission routes between hosts are largely unknown. Nonetheless, avoiding the introduction of non-native species and diseased pollinators appears important to limit spill overs and disease emergence.

## 1. Introduction

Pollinator communities worldwide are declining at an alarming rate [1,2,3]. Emerging viral pathogens have been considered to be major contributors to pollinator losses alongside other drivers, such as habitat destruction; increased use of pesticides and fertilizers; climate change; and biological factors, such as invasive species [1,3,4]. The disruption of pollination services could cause severe effects on modern agriculture and biodiversity [5,6]. Insect pollination accounts for an economic value of over €150 billion annually, and without insect pollinators, food security would be at risk [5]. While managed honey bees and bumble bees are increasingly used to supplement crop pollination, these species cannot replace pollination by wild insects [7].

Viral infections have been extensively studied in the European honey bee (*Apis mellifera*) because a number of RNA viruses that are circulating within bee populations have been associated with reduced health and reports of colony collapses [8,9]. Increasing evidence suggests that many of these pathogens are not specific to honey bees and instead are shared between many pollinator species and associated arthropods, including bumble bees and other wild bees [10,11,12]; bee predators, such as wasps [10,11]; and scavenging insects, such as ants, cockroaches, and beetles [10,13,14,15]. Many of these host species have been found cohabiting in bee hives [13,15,16] or share floral resources with honey bees [17,18,19,20]. Although arthropods harbour an enormous diversity of viruses [21,22], little is known about the host range and dynamics of viruses within pollinator communities.

Between-host virus transmission plays a key role in virus epidemics [23]. Emerging infectious disease are commonly caused by pathogens that infect and interact through multiple host species [24]. Emerging RNA viruses are suspected to spill from honey bees into associates and vice versa [6,19,20,25]. Due to high mutation rates and short generation times, this class of viruses is likely to infect and adapt to new host species and spread through large populations [10,26]. Thus far, transmission routes between species are poorly understood; flowers have been suggested to represent “disease transmission hubs” [27], allowing for transmission without direct interactions among species. Closely related viral strains in several pollinators [10] and clustering of virus strains within geographic regions [12,13,20] indicate that ongoing inter-species virus transmission occurs. In order to improve pollinator conservation, multi-host viruses need to be studied further and outside the *Apis* genus [6].

Species introduction can alter virus communities through the introduction of new pathogens and though introduced hosts providing reservoirs for existing pathogens [28]. For instance, the global spread of the parasitic mite and bee virus vector, *Varroa destructor,* affected virus spread and infection levels in honey bees [29] but also caused pathogen shifts in other pollinators and bee predators, such as wasps [18,30]. Invasive ants with supercolonial social structures in which many interconnected nests form a cooperative network are suspected to tolerate high pathogen loads and form disease reservoirs [31]. In New Zealand, the invasive Argentine ant (*Linepithema humile*) has been hypothesized to form such a reservoir for honey bee viruses [15]. Three bee viruses have been identified in this ant species [15,32] and frequent interactions with honey bees through ants robbing the honey or brood [33] may facilitate virus transmission.

In this study, we tested the hypothesis that a range of pollinators and associated arthropods, including honey bees and Argentine ants, are infected by the same viral pathogens. Furthermore, we hypothesised that frequent inter-species viral transmission results in viral phylogenies that are predominantly associated with geographic origin instead of being associated with host species. Firstly, we screened pollinators and associated arthropods for five RNA viruses and tested for virus replication. Most bee-affecting viruses are positive sense single-stranded RNA (+ssRNA) viruses [34,35], including deformed wing virus (DWV), Kashmir bee virus (KBV), black queen cell virus (BQCV), and Moku virus (MKV). However, we also tested for a negative sense ssRNA (-ssRNA) virus that was recently discovered in Argentine ants, the *Linepithema humile* bunya-like virus 1 (LhuBLV1). Limited knowledge about the host range of these viruses, particularly hosts of ‘bee viruses’ outside bees, makes identifying hosts and potential reservoirs an important step in understanding virus emergence. Secondly, we quantified to what degree the host species and geographic origin of DWV and KBV strains correlate with shared ancestry. We used phylogenetic analysis and phylogeny–trait correlation of our sequences supplemented with viral sequences retrieved from GenBank from various hosts all over the world. Phylogenies separated by geographic regions with hosts interspersed within these can provide evidence for inter-specific transmission. Alternatively, clustering within hosts species could indicate a high degree of host specificity.

## 2. Materials and Methods

### 2.1. Specimen Collections

We used honey bees (*A. mellifera*) and Argentine ants (*L. humile*) as two focal species for this study and sampled these species more frequently than other arthropod species. Adult workers were directly collected from ant nests (*n =* 32) and beehives (*n =* 51) using mouth aspirators or collections jars, respectively. Samples of pollinators and associated arthropods (*n =* 66) were collected using sweep nets or jars when directly collected from abandoned beehives. Collection sites consisted of a mix of agricultural fields, beach edge vegetation strips, and urban areas in the Northland region of New Zealand (Appendix A). A total of 149 samples were collected in April 2018 (*n* = 135) and January 2019 (*n =* 14), some in apiaries, some near Argentine ant nests, and others without an apiary or ant nest nearby (details in Appendix A). Specimens were immediately snap frozen in the field at approximately −150 °C in a liquid nitrogen dry shipper, before being stored at −80 °C in the laboratory until processing. Once in the laboratory, samples were removed from the freezer and placed on ice for identification. When possible, samples were identified to the species level and then again moved to −80 °C until RNA extraction.

### 2.2. RNA Extraction and Reverse Transcription PCR for Virus Detection

Samples were screened for four +ssRNA viruses, DWV, BQCV, KBV, and MKV, and an –ssRNA virus, LhuBLV1, using RT-PCR. MKV has only recently been discovered in wasps and bees [36], and LhuBLV1 in Argentine ants [37]. A preliminary screening of a pooled sample for the seven Argentine ant viruses discovered by Viljakainen et al. [37] concluded that only the –ssRNA virus LhuBLV1 was hosted by insects other than Argentine ants and was therefore included in the analysis. Individual whole arthropod samples were homogenized in 1000 µL (ants in 500 µL) Trizol™ (Life Technologies, Carlsbad, CA, USA) and RNA was extracted and precipitated using isopropanol following the “Trizol™ extraction” protocol as recommended in the COLOSS bee book [38].

RNA was quantified using a NanoDrop™ spectrophotometer and approximately 500 ng of RNA were treated with perfecta DNase (Quanta BioSciences, Gaithersburg, MD, USA) before reverse transcription to cDNA using qScript (Quanta BioSciences, Gaithersburg, MD, USA) following the manufacturer’s instructions. PCR was carried out with MyTaq™ Mix and MyTaq™ Red Mix (Bioline, Meridian Bioscience Inc., London, UK). Here, 15-µL reaction mixes consisted of 1 µL of template (1:20 diluted cDNA) and 0.4 µM of each primer. PCR cycling conditions were 1 min at 95 °C, followed by 35 cycles (38 for KBV) of 15 s of 95 °C, 15 s 55 °C, 30 s 72 °C, and a final 7 min of 72 °C. Positive and negative controls were run alongside every PCR. Primer sequences used in this study can be found in the Appendix A.

RT-PCR products were visualised on a 2% agarose gel (1.5% for DWV) stained with SYBR™ Safe DNA stain (Invitrogen, Life Technologies, Carlsbad, CA, USA) and run alongside a TrackIt 100-bp DNA ladder (Invitrogen, Life Technologies, Carlsbad, CA, USA). A subset of positive samples were cleaned using ExoSAP-IT™ (Applied Biosystems™, Waltham, MA, USA) following the manufacturer’s instructions and sent for Sanger sequencing to Macrogen Inc (Seoul, South Korea). This subset included at least one virus positive sample for every arthropod species for each virus. Sequences were confirmed using a BLASTn search (www.ncbi.nlm.nih.gov/Blast) against the NCBI nucleotide collection and were submitted to GenBank under the accession numbers MT068447 to MT068476.

### 2.3. Negative Strand Detection

DWV, KBV, MKV, and LhuBLV1-positive samples were further analysed for active virus infection. Strand-specific RT-PCR allows for detection of the negative strand, which in +ssRNA viruses (DWV, KBV, MKV) indicates active replication and parasitism of host cells by the virus [39,40]. Similarly, the positive strand in an -ssRNA virus (LhuBLV1) functions as mRNA for protein production or as a template to produce new –ssRNA and indicates an active infection [41]. Equal amounts of RNA from virus-positive samples within each species were pooled and tested for negative or positive strand presence.

Reverse transcription (RT) reactions were carried out in the presence of a tagged primer that contains a non-viral sequence (tag) at the 5′ end of the virus-specific primer [39]. The non-viral sequence is then used in the PCR step in combination with a downstream virus-specific primer, so that only cDNA derived from the strand-specific tagged primer is amplified. Super-Script™ IV First-Strand Synthesis System (Invitrogen, Life Technologies, Carlsbad, CA, USA) was used to reverse transcribe 500 ng of RNA into cDNA following the recommended COLOSS bee book protocol for strand-specific RT-PCR [42]. To avoid the detection of false positives, the remaining RNA and tagged cDNA primers were digested using RNase H (Invitrogen, Life Technologies, Carlsbad, CA, USA) and 10 U Exonuclease-I (Thermo Scientific™, Waltham, MA, USA) at 37 °C for 30 min followed by inactivation at 70 °C for 15 min prior to PCR reactions, as described in the COLOSS bee book [42]. PCR was carried out using tag and reverse primer in a 15-µL reaction using myTaq™ Red Mix (Bioline, Meridian Bioscience Inc., London, UK). Cycling was 1 min at 95 °C, 30 cycles of 15 s at 95 °C, 10 s at 55 °C, 10 to 30 s 72 °C (depending on product length), and a final 5 min of 72 °C. The low cycling number was used to reduce the occurrence of unspecific bands. PCR products were visualised on an agarose gel and sequenced to confirm products as described above. 

### 2.4. Statistical Analyses

Statistical analyses were performed in R v3.4.1 [43]. Virus prevalence with 95% confidence intervals in the two focal species was determined using the *epiR* package [44] with 95% sensitivity and specificity. Using the *prop.test()* function within R (R Foundation for Statistical Computing, Vienna, Austria, [43]), we tested if proportions of infected individuals differed between Argentine ants and honey bees. 

### 2.5. Phylogenetic Analyses

To conduct an analysis of globally occurring viral variants in different host species, we supplemented DWV and KBV sequences generated in this study with sequences obtained from GenBank (www.ncbi.nlm.nih.gov/genbank/). We used the same genomic regions that were used in the virus detection assay to conduct the phylogenetic analyses, which were a fragment of the RNA-dependent RNA polymerase (*RdRp)* for DWV and the major capsid protein *vp3* for KBV. Through a BLASTn search, we retrieved 160 sequences that matched the DWV *RdRp* fragment. Only DWV-A was included in the analysis because all DWV sequences from our sample in New Zealand matched the DWV-A group. For KBV, 11 sequences matching the *vp3* fragment were found and added to the sequences from this study. Most DWV and KBV sequences available on GenBank are from *A. mellifera* hosts; therefore, we tested an additional 34 Argentine ant samples obtained by Felden et al. (2019) from Argentina, California, and France (details in Appendix A) [45] for DWV and KBV. Briefly, RNA was extracted from single ants using the Direct-zol RNA Microprep extraction kit (Zymo Research) following the manufacturer’s instructions. Approximately 50 ng of RNA were reverse transcribed using the High-Capacity cDNA Reverse Transcription kit (Applied Biosystems™), tested for DWV and KBV, and sequenced using the methods described above. From these samples, two DWV and nine KBV sequences were obtained and included in the analysis.

Sequences were trimmed and aligned using the ClustalW algorithm [46] in Geneious v11.1.5 [47]. MEGA v.10.1 [48] was used to find the best substitution model using Bayesian information criterion (BIC) scores, determining HKY+G as the best model for both viruses. Final alignments were 440 bp in the *RdRp* region for DWV and 360 bp in the *vp3* region for KBV.

Dated Bayesian phylogenetic analyses were performed in BEAST 2.2.6 [49]. Divergence times were calculated based on a tip-dated coalescent model. Clade probabilities were obtained from the posterior distribution incorporation sampling year information (retrieved from GenBank or original publications) for terminal nodes. Data were run under an uncorrelated lognormal relaxed clock model with gamma-distributed rate heterogeneity and coalescent tree prior with exponential population growth. Bayesian analyses were replicated three times and combined, each with Markov chain Monte Carlo (MCMC) of 150 million generations. Trees were sampled every 10,000 generations, of which the first 10% were discarded as burn-in. Using TreeAnnotator v1.7.5, we constructed maximum clade credibility (MCC) trees and used FigTree v1.4.4 to visualise trees, including high posterior probabilities over 0.6. Tip-dated phylogenies are commonly used to reconstruct RNA viral evolution [50,51]; however, there was a chance that the sampling date and sampling region in our dataset would be connected. Therefore, we repeated the analysis without using the tip-date calibration and compared dated and non-dated phylogenies.

### 2.6. Phylogeny–Trait Correlation

Phylogeny–trait association analysis was used to test to which extent phenotypic traits of a viral strain, such as host species or geographic location, are correlated with shared ancestry [52]. Association sndex (AI), parsimony score (PS) [53], and monophyletic clade (MC) scores of dated and non-dated phylogenies were computed using BaTS (Bayesian tip-association significance testing) beta build 2 [52] by comparing a null model of random trait–tip assignments to the known trait distribution. BaTS accounts for phylogenetic uncertainty by testing many trees from the posterior distribution. Trait–tip association analyses were run with 500 replicates (500 random trees from the dated Bayesian phylogenetic analysis) to estimate a null distribution for each statistic.

## 3. Results

### 3.1. Viral Prevalence in Pollinators and Associated Arthropods

We collected arthropods from eight different orders. Virus-positive species were found in all except the Odonata (Table 1), but we note that only two samples were taken from this order. Overall, viral infections were common in pollinators and associates, with 83% (95% CI: 76%–88%) of samples testing positive for at least one virus and 44% (95% CI: 36%–53%) having multiple infections. The majority of samples were hymenopterans (*n* = 105), out of which 97% (95% CI: 92%–99%) were infected with at least one and many with multiple (62%, 95% CI: 52%–73%) viruses. One individual had as many as four viruses. Outside the order Hymenoptera (*n* = 46), 51% (95% CI: 37%–65%) of samples were found positive for at least one virus but only 6% (95% CI: 2%–17%) for multiple (two) viruses.

### 3.2. Viral Coinfections in Honey Bees and Argentine Ants

Out of the five viruses tested, up to four were found to co-infect individual workers of Argentine ants and honey bees (Figure 1). The most prevalent viruses were DWV in honey bees (100%, 95% CI: 94%–100%) and LhuBLV1 in Argentine ants (78%, 95% CI: 61%–90%) (Table 2). LhuBLV1 was only detected in Argentine ants and associated arthropods in two sites that had ant nests, indicating that other species can host this virus, or that its detection on other arthropods could have resulted from contamination. Only Argentine ants tested positive for the mRNA intermediate of LhuBLV1, suggesting that this virus may be *L. humile* specific. Interestingly, all ants that tested positive for DWV were sampled from a nest in an apiary (Appendix A, site WAI). In a test of proportions, DWV was more prevalent in bees than in ants (Χ^2^ = 35.98, *p* < 0.001), KBV was more prevalent in ants than in bees (Χ^2^ = 6.69, *p* = 0.010), and MKV infections were not significantly different between the two species (Χ^2^ = 0.15, *p* = 0.689) (Table 2). MKV was found at a high prevalence in Vespinae species (n= 17, 88% infected, 95% CI: 65%–98%); this group also showed a KBV prevalence similar to honey bees (24%, 95% CI: 8%–49%).

### 3.3. Viral Replication within Host Species

All viruses, except BQCV which was only detected in honey bees, were tested for virus replication, or active virus infections in the case of LhuBLV1. Active LhuBLV1 infection was only found in Argentine ants while all +ssRNA viruses replicated in multiple species (Table 1). Fourteen different species tested positive for DWV; active viral replication could be confirmed for five of them (Table 1). For KBV, 5 out of the 9 virus positive species showed active replication and MKV replication was confirmed in 5 out of 10 species including the *Polistes* and *Vespula* species tested (Table 1). The negative strand RT-PCR assay is highly conservative and can only confirm the presence of the negative strand intermediate, which is indicative of viral replication but does not allow the exclusion of the possibility for replication.

### 3.4. Viral Strain Diversity and Phylogenetic Analysis

Using BLAST searches, we confirmed the virus identity and identified locations and hosts infected with viral strains most similar to those observed in our samples. BLAST searches showed that BQCV most closely matched the polyprotein of BQCV found in a honey bee in Lithuania (KP223790) (96% identity, 100% query cover). The MKV sequences from 10 different species showed little sequence variation; all sequences derived in this study closely matched (99%–100% identity, 97%–100% query cover) the MKV polyprotein found in *Vespa velutina* in Belgium (MF346349) and *Vespula pensylvanica* from Hawaii (KU645789). LhuBLV1 most closely matched the putative *RdRp* complex gene of the only available LhuBLV1 sequences on GenBank (MH213237) from Argentine ants from Spain (100% identity, 95% query cover).

The phylogenetic analysis showed that DWV sequences found in New Zealand in this study clustered within the DWV-A clade and formed a monophyletic group with a sequence from *A. mellifera* from New Zealand (MF623172) and *Vespa carbo* from Italy (KY909333) (posterior probability (pp) = 0.539, Figure 2). This group is dated to the year 2013 (95% highest posterior density (hdp) 2010–2016). However, when not including the sampling date in the phylogeny, sequences derived in this study clustered with a broader group of European DWV sequences (Appendix A). Two DWV strains from Pakistan (KP734706, KP734705) represented a relatively old clade (pp = 1) that was estimated to have separated from other DWV sequences around 1977 (95% hdp = 1965–1989) (Figure 2). The DWV phylogeny further split into two sister groups (pp = 0.95, dated to 1986, 95% hdp = 1979–1994), one from Asia and Europe, which includes the DWV-A variant Kakugo virus, and the other with strains from Europe, North and South America, and New Zealand (Figure 2 and Appendix A, non-dated phylogeny). Within this group, samples from New Zealand all occupy a different clade than samples from the Americas, including Hawaii, and both share these with Europe (pp = 1, dated to 1999, 95% hdp = 1998–2001, Figure 2 and Appendix A, non-dated phylogeny). Interestingly the two strains collected from Australia (KP734699, KP734632) did not cluster together or with strains from New Zealand.

The phylogenetic analysis for KBV showed that the sister group of most New Zealand KBV sequences was a group from Tasmania (pp = 0.641, Figure 3 and Appendix A, non-dated phylogeny). KBV from a New Zealand *V. vulgaris*, however, closely matched samples from the USA (HM228885, HM228887) (pp = 1, Figure 3 and Appendix A, non-dated phylogeny).

### 3.5. Trait–Tip Associations

The trait–tip association analysis for DWV showed significant clustering within geographic locations and within host species in both dated and non-dated phylogenies (association index (AI) and parsimony score (PS), *p* < 0.01, Table 3, and Appendix A, trait–tip association in non-dated phylogeny). The PS, which indicates the number of state changes on a tree, was smaller for geographic locations than host species (Table 3 and Appendix A). Overall observed to expected ratios indicated a stronger association with geographic origin than with host species (Table 3, Appendix A). The maximum monophyletic clade (MC) index tests which traits are associated with phylogeny, and showed that the maximum observed clade size for Asia, Europe, North America, and Oceania was larger than expected by chance (all MC scores *p* < 0.01, Table 3 and Appendix A). The association was also statistically supported (all MC scores *p* < 0.05) for a number of host states that included *Apis, Varroa*, bee associates, non-*Apis* bees, and *L. humile* (Table 3, bee associates *p* = 1 in non-dated analysis, Appendix A), which indicates that these groups carry strains that are more closely related than expected due to chance. Yet, it is difficult to determine whether a geographic state or host state causes the association as geographic origin and host species were often linked. For example, all North American samples in the analysis were from *A. mellifera* hosts and all bee associates were collected in New Zealand. Therefore, host and geographic origin can both contribute to an observed association.

For KBV, significant clustering within geographic regions and within host species was found (AI and PS, *p* < 0.01, Table 4 and Appendix A, trait–tip association in non-dated phylogeny). Again, a stronger association with geographic origin was observed than with host species (observed to expected ratio, Table 4). Due to the limited number of available KBV sequences covering the *vp3* region (*n* = 24), samples from a geographic location were often sourced from the same host, and vice versa. KBV sequences significantly clustered within New Zealand, Europe, and Australia (all MC: *p* < 0.05) but also within bumble bees (MC: *p* < 0.01) (Table 4 and Appendix A). The 10 *L. humile* samples that were positive for KBV did not cluster together (MC: *p* = 0.16, Table 4) and instead sequences appeared to be divided by geographic region.

## 4. Discussion

Our study shows that multiple RNA viruses that were initially detected in honey bees or Argentine ants infect a range of pollinators and associated arthropods. We have added to the growing body of evidence of bee viruses infecting and replicating in non-*Apis* hosts. To our knowledge, this work represents the first study to confirm active LhuBLV1 and MKV infections in arthropod hosts by detecting the positive and negative viral strand intermediate, respectively. Virus replication in novel hosts is a key contributor in disease emergence; nevertheless, even hosts without active virus replication may show active virus replication at a different time [54] or can be contaminated with infectious viral particles that contribute to disease spread [6]. The arthropods we examined typically did not display any symptoms of viral infection. The absence of obvious symptoms is common in dicistroviruses like BQCV and KBV [55], yet there may be negative effects on the host. For example, covert KBV infections in bumble bees reduce reproduction [56] and DWV-infected honey bees rarely display wing deformities, although covert infections reduce foraging and long-term survival [20,57]. To which extent these viruses affect hosts other than bees is not well characterised and remains to be investigated [6].

### 4.1. Host Range of RNA Viruses

DWV appears to be the most prevalent honey bee pathogen, occurring in approximately 55% of colonies worldwide [58]. We found high DWV prevalence in honey bees but also in Argentine ants. The list of DWV hosts is long [58]. We add to it by reporting the first DWV case in the order Orthoptera (*Bobilla*) and in New Zealand native Hymenoptera (*Xanthocryptus novozealandicus* and *Sphictostethus nitidus*), Coleoptera (*Scolopterus penicillatus*), and Blattodea (*Maoriblatta*).

In case of the more recently discovered viruses, such as MKV and LhuBLV1, the knowledge of disease symptoms, dynamics, or host range is very limited [36,37]. Studies suggest that the predatory wasp *V. pensylvanica* may represent an MKV reservoir on Hawaii [16,36]. We confirmed active MKV replication in three different wasp species and an overall high prevalence in Vespinae, supporting the potential virus reservoir. Negative sense ssRNA viruses of the Bunyavirales family are found in many species, including LhuBLV1 in Argentine ants [37] and *Apis mellifera* Bunyavirus 1 and 2 in honey bees [59]. We detected the positive strand intermediate of LhuBLV1 in Argentine ants but not in any other host. LhuBLV1 did not seem specific to Argentine ants, as four other species were found virus positive. Yet, no pollinator species tested positive for this virus, making LhuBLV1 unlikely to be an emerging virus that poses a risk on pollinator communities.

### 4.2. Pathogen Reservoir and Viral Spill Over into Wild Populations

Pathogen transmission may potentially alter competition among multiple host species so that spatial interactions are important when regarding transmission networks [60]. RNA viruses initially identified in honey bees can be widespread in the environment [27], but only some novel hosts acquire infections. High densities of apiaries or introduced ants, such as the Argentine ant, could allow accumulation of high viral levels and possible spill over into pollinators and associates that would account for the virus-positive species found in this study. Frequently, the main drivers for disease emergence in wildlife populations are spill overs from domesticated hosts and human-mediated pathogen invasion, also termed ‘pathogen pollution’ [61]. Introduced pathogens can provide a competitive benefit to introduced hosts if native hosts are more susceptible to the pathogen than the invader [28]. Moreover, domestic animals often outnumber wild species and can act as reservoir hosts that may drive population decline of the wild host [61]. For instance, the introduction of rabies virus into the Serengeti ecosystem through domestic dogs caused spill overs into wild carnivore communities and ongoing efforts to vaccinate dogs are required to stop the disease spread [62]. In case of the European honey bee, *Varroa*-mediated high DWV titres in bees may cause spill overs and promote virus spread to associates and other pollinators [63]. The presence of apiaries has shown to cause a high prevalence of DWV and BQCV in wild pollinators [19] and species associated with beehives [10,13,64]. We found BQCV in honey bees but no other pollinator, which may be due to the higher BQCV prevalence in honey bees compared to other bees [65], and the low number of wild bee pollinators in our sample set. We only detected DWV in Argentine ants from apiaries, which could indicate that ants acquire the virus from bees, potentially by scavenging in apiaries. In addition to species introduction, anthropogenic actions reducing biodiversity in landscapes and species may facilitate disease transmission and outbreaks [66].

### 4.3. Coinfections and Interactions among Viruses

Coinfections and interactions between viral species within the host can play a key factor in the epidemiology and evolution of viruses [67]. Many insects that we analysed tested positive for multiple viruses. Interactions within hosts and, for example, the order in which pathogens are acquired may have a large positive or negative effect on a secondary infection [68,69]. In some cases, infection with one virus can prevent infection with another: Consequently, this makes the infection beneficial if it is less damaging or virulent than the other [37,70]. Multiple parasites may compete for resources or even release toxins that inhibit the growth of a competitor [71,72]. These interactions affect adaptive responses to the infection and dynamics within the community [71,73]. Thereby interactions can be strain specific and affect parasite evolution and genetic diversity within virus populations [74]. Future work is needed to understand co-infections and interactions among viruses in addition to the commonly studied host–parasite interactions [68].

### 4.4. Multi-Host Viruses in Emerging Disease

Single-stranded RNA viruses have been considered to be the most likely type of pathogen to jump between species and cause disease outbreaks [23] that could devastate wild pollinator populations. High mutation rates in RNA viruses result in virus populations with high variation amongst genotypes [26]. Over time, multi-host parasites can evolve into different species-specialist strains or populations, or remain a generalist parasite with lower than optimal virulence [75]. For multi-host viruses, a lack of host specificity may result in a trade-off between virulence in different hosts that depends on the host quality and availability [75]. Whether a virus that invades a new host causes an emerging disease or only a minor outbreak depends factors, such as its ability to replicate in a novel host and transmission routes [6,23].

We confirmed DWV, KBV, and MKV replication in a number of species, indicating that these species possibly act as biological vectors that facilitate disease emergence. However, with pollinators and associates collected in the field, we cannot exclude the possibility of detecting viruses and negative strand intermediates in the gut contents that come from other infected insects. KBV and MKV replication was confirmed in species not typically known to eat bees, such as *Polistes* wasps and bumble bees. Some cockroaches, crickets, wasps, and ants that tested positive for virus replication were found in close proximity to beehives and are species known to scavenge or prey on honey bees [30,33]. Recently consumed viral particles from infected honey bees can cause false positives in replication assays. For instance, *Varroa* mites that exclusively feed on honey bee tissue [76] and vector DWV are assumed to act as a biological vector that propagates DWV [39,77]. Yet, new research suggests that DWV may not replicate in mite cells but in honey bee cells recently consumed by mites [78]. Nevertheless, feeding experiments have shown that scavenger species like the ant *Myrmica rubra* are found positive for the negative strand of DWV for up to 13 weeks after consuming infected honey bees [14], which indicates that DWV actively infects ants. Whether honey bee scavengers are able to spread viruses without becoming infected themselves [79] or aid in reducing virus transmission by removing infectious carcasses [80] remains to be tested. The detection of negative strand intermediates is an essential step in identifying host species, but only controlled infection experiments allow determination of the virus dynamics in these hosts and the potential for disease emergence.

### 4.5. Global Distribution and Evolution of Bee Viruses

To understand the current epidemic of bee viruses, it is key to determine the role of the host species and geographic distribution [12,65]. By placing the sequences generated in this study within a global phylogeny, we found that both KBV and DWV showed strong grouping by geographic regions. Closely related viral strains in geographic locations support the hypothesis that viral infections can be acquired from the environment and other host species in the same location [12]. Although DWV was present in New Zealand before the introduction of *Varroa* in 2001 [81], all New Zealand DWV strains grouped within the clade that emerged in the early 2000s. The DWV phylogeny showed some broad separations into geographic regions, and only European samples were scattered across groups. Whether European strains are found across the phylogeny because introduced honey bees or other virus-carrying arthropods from Europe initiated the spread remains to be tested further. Research indicates that the global DWV epidemic has been mediated by European *A. mellifera* populations but that the virus shows little host specificity [63]. Although the KBV phylogeny only represents a limited number of the true KBV diversity, sequences from Argentine ants showed weak association with the phylogeny, suggesting that ants acquired local KBV strains. Moreover, the high prevalence of KBV in this species could indicate a disease reservoir, and ants facilitating further spread of KBV.

## 5. Conclusions

Many emerging viruses infect and actively replicate within a wide range of pollinators and associated Arthropod species. Our results and others indicate that the host species can affect the association with shared ancestry, which indicates virus transmission, but that these effects are weaker than the geographic location [13]. Overall, geographic locations played a major role in shaping patterns of viral genetic diversity. As viruses may be frequently transmitted between species, it is important to incorporate the host abundance and diversity, spatial structure of communities, and multi-host systems into studies of pathogen dynamics and epidemics [60]. Avoiding the introduction of non-native species and transport of diseased pollinators is important to stop disease spill overs and prevent disease emergence.

## Figures and Tables

**Figure 1 viruses-12-00358-f001:**
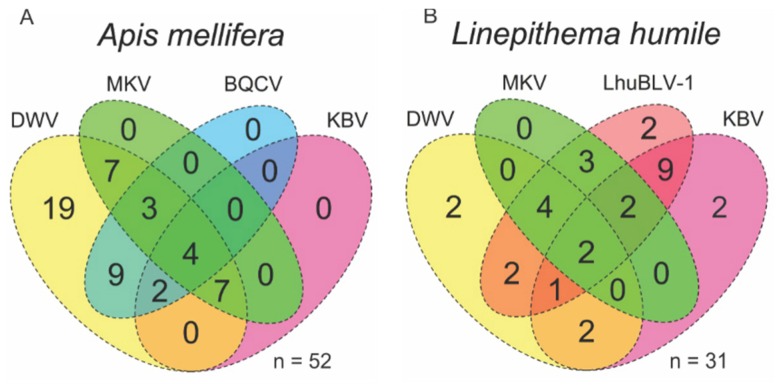
Venn diagram of virus coinfection in honey bees (*Apis mellifera*) (**A**) and Argentine ants (*Linepithema humile*) (**B**). Most prevalent viruses were DWV in honey bees and LhuBLV1 in Argentine ants. BQCV: black queen cell virus; DWV: deformed wing virus; KBV: Kashmir bee virus; LhuBLV1: Linepithema humile bunya-like virus 1; MKV: Moku virus.

**Figure 2 viruses-12-00358-f002:**
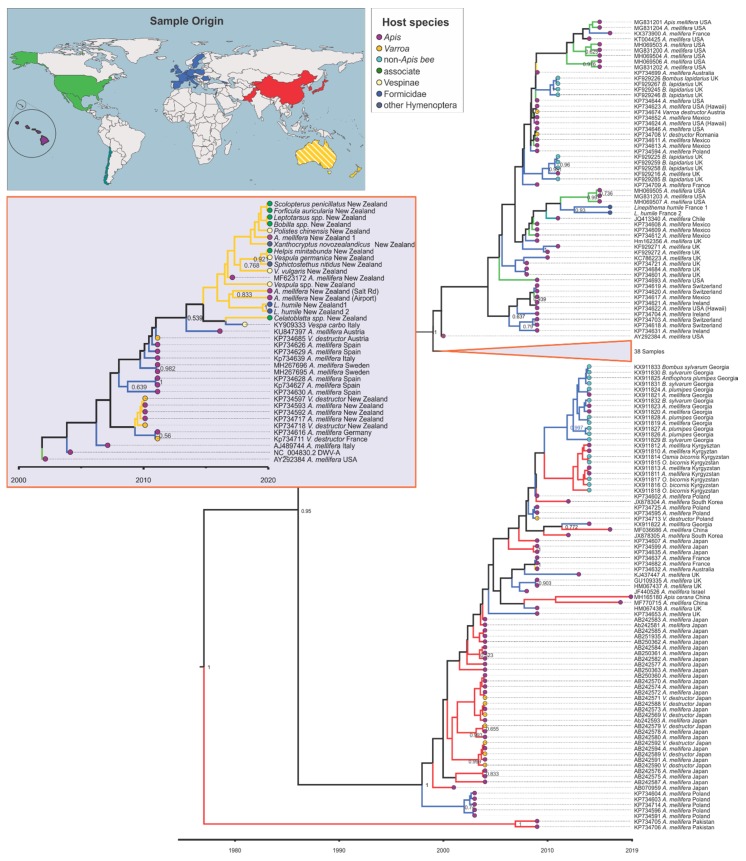
Maximum clade credibility tree for the RNA-dependent RNA polymerase (*RdRp*) fragment (440 bp) of deformed wing virus (DWV). Grey insert with orange frame shows the collapsed part of the tree that among others includes the 19 samples from New Zealand from this study. Species name, country of origin, and, if applicable, GenBank accession number are given in the branch label. The branches are coloured according to the lineages’ inferred geographic origin as shown on the world map and end notes are coloured according to the host group as shown in the host species insert. Posterior support > 0.6 is given and the x-axis shows time in years.

**Figure 3 viruses-12-00358-f003:**
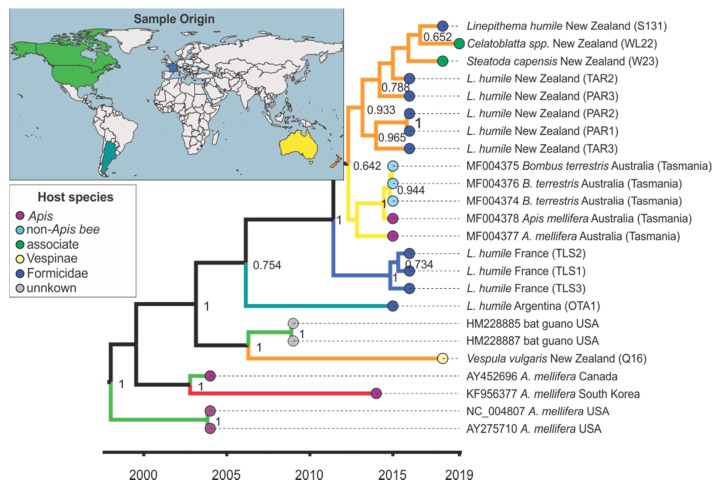
Maximum clade credibility tree for the major capsid protein *vp3* fragment (360 bp) of Kashmir bee virus (KBV). Species name, country of origin, and, if applicable, GenBank accession number are given in the branch label. The branches are coloured according to the lineages’ inferred geographic origin as shown on the world map and end notes are coloured according to the host group as shown in the host species insert. Posterior support > 0.6 is given and the x-axis shows time in years.

**Table 1 viruses-12-00358-t001:** Virus presence in pollinators and associated arthropods tested for deformed wing virus (DWV), Kashmir bee virus (KBV), Moku virus (MKV), black queen cell virus (BQCV), and Linepithema humile bunya-like virus 1 (LhuBLV1) using RT PCR. Virus not found in species: -; virus found in species: +; active viral replication confirmed: +/+. Summary in bottom line.

Order	Family	Genus	Species	Common Name	*n*	DWV	KBV	MKV	BQCV	LhuBLV1
Araneae	Salticidae	*Helpis*	*Helpis minitabunda*	jumping spider	1	+	-	+	-	+
	Lycosidae	*Lycosa*	*-*	wolf spider	1	-	-	-	-	-
	Theridiidae	*Steatoda*	*Steatoda capensis*	black cobweb spider	3	-	+	+/+	-	-
	Thomisidae	*Diaea*		flower spider	4	-	-	-	-	-
Blattodea	Blattidae	*Celatoblatta*	*-*	cockroach	9	+/+	+	+	-	-
		*Maoriblatta*	*-*	cockroach	7	+	-	+	-	+
Coleoptera	Curculionidae	*Scolopterus*	*Scolopterus penicillatus*	black spined weevil	1	+	-	-	-	-
Diptera	Sarcophagidae	*Jantia*	*Jantia crassipalpis*	flesh fly	2	-	-	-	-	-
	Stratiomyidae	*-*	*-*	solider fly	2	-	-	-	-	+
	Tipulidae	*Leptotarsus*	*-*	crane fly	2	+	-	-	-	-
Dermaptera	Forficulidae	*Forficula*	*Forficula auricularia*	European earwig	2	+	+	-	-	-
Hymenoptera	Apidae	*Apis*	*Apis mellifera*	honey bee	51	+/+	+/+	+	+	-
		*Bombus*	*-*	bumble bee	2	-	+/+	-	-	-
	Formicidae	*Linepithema*	*Linepithema humile*	Argentine ant	32	+/+	+/+	+/+	-	+/+
	Pompilidae	*Sphictostethus*	*Sphictostethus nitidus*	golden hunter wasp	1	+	-	-	-	-
	Vespidae	*Polistes*	*Polistes chinensis*	Chinese paper wasp	11	+	-	+/+	-	-
			*Polistes humilis*	Australian paper wasp	2	-	-	+	-	-
		*Vespula*	*Vespula germanica*	German wasp	1	+/+	+	+/+	-	-
			*Vespula vulgaris*	common wasp	3	+/+	+/+	+/+	-	-
	Ichneumonidae	*Xanthocryptus*	*Xanthocryptus novozealandicus*	lemon tree borer parasite	1	+	-	-	-	-
Odonata	Lestidae	*Austrolestes*	*Austrolestes colensonis*	blue damselfly	2	-	-	-	-	-
Orthoptera	Acrididae	*Locusta*	*Locusta migratoria*	migratory locust	1	-	-	-	-	-
	Gryllidae	*Bobilla*	*-*	small field cricket	6	+	+/+	-	-	+
		*Teleogryllus*	*Teleogryllus commodus*	black field cricket	4	-	-	-	-	-
Total virus-positive species (samples)					21 (151)	14 (83)	9 (44)	10 (53)	1 (18)	5 (32)

**Table 2 viruses-12-00358-t002:** Estimated viral prevalence in honey bees (*Apis mellifera*) and Argentine ants (*Linepithema humile*) for five virus targets with 95% confidence intervals (CI) in brackets. - indicates that the virus was not detected in the species and * indicates a significantly higher prevalence in a test of proportions. BQCV: black queen cell virus; DWV: deformed wing virus; KBV: Kashmir bee virus; LhuBLV1: Linepithema humile bunya-like virus 1; MKV: Moku virus.

Host	BQCV	DWV	KBV	LhuBLV1	MKV
*A. mellifera*	35% (23%–49%)	100%* (94%–100%)	25% (15%–39%)	-	41% (28%–55%)
*L. humile*	-	41% (24%–58%)	56%* (39%–72%)	78% (61%–90%)	34% (20%–53%)

**Table 3 viruses-12-00358-t003:** Trait–tip association in deformed wing virus (DWV), as calculated with 500 random trees (replicates) in BaTS (Bayesian tip-association significance testing) [52]. Significant *p* values are in bold font. Low parsimony score (PS) represents fewer state changes on the tree and stronger trait–tip association. Maximum clade (MC) scores show the maximum size of a clade for a trait state; a high MC is positively correlated with trait–tip association. The geographic region of South America was only represented by one sample and is not shown.

Statistic	n	Observed to Expected Ratio (95% CI)	Observed Mean (95% CI)	Null Mean (95% CI)	*p* Value
*Association Index* (*AI*)					
Geographic location	-	0.21 (0.13–0.30)	2.83 (1.95–3.74)	13.43 (12.47–14.47)	**<0.01**
Host species	-	0.55 (0.40–0.74)	4.96 (3.89–6.01)	8.98 (8.17–9.76)	**<0.01**
*Parsimony Score* (*PS*)					
Geographic location	-	0.35 (0.30–0.40)	30.91 (28.00–34.00)	88.56 (84.43–92.45)	**<0.01**
Host species	-	0.68 (0.61–0.74)	34.87 (32.00–37.00)	51.30 (49.75–52.35)	**<0.01**
*Maximum Clade* (*MC*) *scores*					
Asia	50	-	15.59 (11.00–21.00)	2.57 (2.12–3.27)	**<0.01**
Europe	76	-	14.44 (14.00–16.00)	3.53 (2.89–4.79)	**<0.01**
North America	24	-	4.48 (3.00–8.00)	1.66 (1.28–2.09)	**<0.01**
Oceania	25	-	12.54 (6.00–18.00)	1.72 (1.29–2.24)	**<0.01**
Hawaii	3	-	1.07 (1.00–2.00)	1.01 (1.00–1.06)	1
*Apis*	126	-	13.24 (9.00–21.00)	7.04 (5.37–9.93)	**0.01**
*Varroa destructor*	13	-	1.70 (1.00–3.00)	1.27 (1.01–1.99)	**0.04**
*Associate*	6	-	1.83 (1.00–3.00)	1.05 (1.00–1.19)	**<0.01**
Non*-Apis* bee	23	-	4.23 (4.00–5.00)	1.63 (1.23–2.07)	**<0.01**
*Vespidae*	5	-	1.09 (1.00–2.00)	1.03 (1.00–1.15)	1
*Formicidae*	4	-	2.00 (2.00–2.00)	1.02 (1.00–1.07)	**<0.01**
Other Hymenoptera	2	-	1.00 (1.00–1.00)	1.00 (1.00–1.00)	1

**Table 4 viruses-12-00358-t004:** Trait–tip associations for Kashmir bee virus (KBV), as calculated with 500 random trees (replicates) in BaTS (Bayesian tip-association significance testing) [52]. Significant *p* values are in bold font. Low parsimony score represents fewer state changes on the tree and stronger trait-tip association. Maximum clade scores show the maximum size of a clade for a trait state; a high MC is positively correlated with trait–tip association. The geographic regions of Asia of South America were only represented by one sample each and are not shown.

Statistic	n	Observed to Expected Ratio (95% CI)	Observed Mean (95%CI)	Null Mean (95%CI)	*p* Value
*Association Index* (*AI*)					
Geographic location		0.16 (0.13–0.25)	0.36 (0.33–0.43)	2.19 (1.74–2.63)	**<0.01**
Host species		0.27 (0.11–0.50)	0.58 (0.29–0.79)	2.14 (1.59–2.57)	**<0.01**
*Parsimony Score* (*PS*)					
Geographic location		0.45 (0.40–0.58)	6.06 (6.00–7.00)	13.44 (12.12–14.92)	**<0.01**
Host species		0.59 (0.50–0.73)	7.55 (7.00–8.00)	12.74 (11.00–13.98)	**<0.01**
*Maximum Clade* (*MC*) *scores*					
North America	4	-	2.00 (2.00–2.00)	1.26 (1.00–2.00)	0.10
New Zealand	9	-	7.83 (6.00–7.00)	1.92 (1.07–3.00)	**0.01**
Europe	3	-	3.00 (3.00–3.00)	1.10 (1.00–2.00)	**0.01**
Australia	5	-	4.37 (4.00–5.00)	1.24 (1.00–2.00)	**0.01**
*Apis*	6	-	2.54 (2.00–4.00)	1.45 (1.00–2.94)	0.19
*Bombus*	3	-	2.97 (2.00–3.00)	1.10 (1.00–2.00)	**<0.01**
*Linepithema*	10	-	3.08 (3.00–4.00)	2.12 (1.10–3.44)	0.16
associate	2	-	1.09 (1.00–2.00)	1.03 (1.00–1.17)	**1**

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
