# Peer review of "Genetic Strain Diversity of Multi-Host RNA Viruses that Infect a Wide Range of Pollinators and Associates is Shaped by Geographic Origins"

_viruses, 2020, doi:10.3390/v12030358_

Round 1

Reviewer 1 Report

The manuscript provides important new information on host range of natural insect viruses by showing that viruses transmit from species to species locally as the virus phylogenies are associated with geographic regions rather than species.

There are some changes that I feel could improve and clarify the manuscript.

Major points:

  1. You should explain in the text why did you choose to test four known bee infecting picornaviruses (DWV, BQCV, KBV, Moku) and compare those with Argentine ant infecting bunyavirus with completely different genome structure: negative strand and segmented? This makes sample analyzing unequal, for example, the replication test is not the same since you do not measure replication of LhuBLV1. Does the manuscript need the LhuBLV1 results?
  2. A virus name should not be italicized (see ictvonline.org/information/w/faq/386/how-to-write-virus-and-species-names)
  3. Very little of the manuscript results are discussed in the “Discussion” section, for example, parts 4.2 and 4.4. do not even mention any results from the manuscript. Are these sections needed or could some of the results be integrated into them? For example, could you discuss why you results do not show high prevalence of BQCV in other species?

Minor points:

Abstract

  1. Line 24: Is “future work” at right place in the abstract?

Introduction

  1. Lines 32-35: Do the referenced articles show that any pollinator decline is due to viruses? Or that viruses are even a major contributor?
  2. Lines 43-44 and line 50: could examples be given about pollinators and associated arthropods?
  3. Line 60: “mite Varroa destructor that vectors bee viruses has not only changed viral disease in honey bees“ Could this sentence be explained? You mean that DWV was not disease causing before, but become such due to the mite?

Materials and Methods

  1. Line 88: Missing a word?
  2. “2.5 Phylogenetic analysis”: Please, explain why phylogenetic analysis was done with RdRP for DWV but VP2 for KBV?

Results

  1. Lines 194-195, 203 and 325: If LhuBLV1 can be found from other arthropods but it cannot be shown to produce mRNA, the virus might still be specific to Argentine ant and be just a contamination in other insects. A virus that does not produce mRNA, does not productively replicate either.
  2. Table 1.: Common names: capital letter or not. Soldier fly?
  3. Table2.: Species?
  4. “3.4 Viral strain diversity and phylogenetic analysis”: Blast search results should be somewhere to see and “closely matched” should be clarified -what does it mean in numbers? Were complete genome sequences or RdRP/capsid “Blasted” for BQCV, moku and LhuBLV1? (methods?) Was there really only one matching sequence for BQCV?

Discussion

  1. Line 301: It is not shown in this manuscript that these viruses cause harm to the host and thus parasite should be removed.
  2. Line 303: It was not shown that LhuBLV1 is replicating because there was not such an assay. You showed that the virus produces mRNA.
  3. Line 306. “Or regulatory mechanisms may facilitate viral replication at a different time”. Clarify/elaborate this sentence.
  4. Line 310: although.
  5. Line 358: are.

Author Response

Reviewer 1:

The manuscript provides important new information on host range of natural insect viruses by showing that viruses transmit from species to species locally as the virus phylogenies are associated with geographic regions rather than species.

There are some changes that I feel could improve and clarify the manuscript.

>>> We thank the reviewer for their thoughtful and relevant comments.

Major points:

  1. You should explain in the text why did you choose to test four known bee infecting picornaviruses (DWV, BQCV, KBV, Moku) and compare those with Argentine ant infecting bunyavirus with completely different genome structure: negative strand and segmented? This makes sample analyzing unequal, for example, the replication test is not the same since you do not measure replication of LhuBLV1. Does the manuscript need the LhuBLV1 results?

>>> In a preliminary screening for Argentine ant viruses on a pooled sample consisting of RNA from all non-honey bee samples in our pool and all honey bee samples we tested for seven Argentine ant viruses that were discovered by Viljakainen et al., 2018 detected only LhuBLV1 outside of Argentine ants. We have modified L105-107 and included a sentence explaining why we chose to include LhuBLV1 in the analysis. “A preliminary screening of a pooled sample for the seven Argentine ant viruses discovered by Viljakainen et al. [37] concluded that only the –ssRNA virus LhuBLV1 hosted by insects other than Argentine ants and therefore included in the analysis.” L105-107

  1. A virus name should not be italicized (see ictvonline.org/information/w/faq/386/how-to-write-virus-and-species-names)

>>> We have changed this, virus names are no longer italicised.

  1. Very little of the manuscript results are discussed in the “Discussion” section, for example, parts 4.2 and 4.4. do not even mention any results from the manuscript. Are these sections needed or could some of the results be integrated into them? For example, could you discuss why you results do not show high prevalence of BQCV in other species?

>>> We have modified section 4.2 of the manuscript to link to Argentine ants as a potential species that facilitates virus spill-overs and link to our results at the end of the paragraph. We also briefly mention the finding that BQCV was only present in honey bees and suggest that low number of wild bee pollinators sampled may have caused this finding. The modified or added sentences are the following:

>>> “High densities of apiaries or introduced ants, such as the Argentine ant, could allow accumulation of high viral levels and possible spill-over into pollinators and associates that would account for virus positive species found in this study.” L359-361

>>> “We found BQCV in honey bees but no other pollinator, which may be due to higher BQCV prevalence in honey bees compared to other bees [65], and low number of wild bee pollinators in our sample set.” L378-380

>>> “We only detected DWV in Argentine ants from apiaries, which could indicate that ants acquire the virus from bees, potentially by scavenging in apiaries.” L380-381

Minor points:

Abstract

  1. Line 24: Is “future work” at right place in the abstract?

>>> This sentence has been reworded to avoid “future work” and now reads: “Transmission routes between hosts are largely unknown, nonetheless avoiding introduction of non-native species and diseased pollinators appears important to limit spill-overs and disease emergence.” L24-26

Introduction

  1. Lines 32-35: Do the referenced articles show that any pollinator decline is due to viruses? Or that viruses are even a major contributor?

>>> We have reworded the first two sentences of the introduction. The first sentence now only refers to the alarming rate of pollinator declines, while the second sentence mentions viruses as a driver of declines. The references cited (Potts et al., 2010 [1], Sánchez-Bayo & Wyckhuys 2019 [3], and Goulson et al., 2015 [4]) all discuss parasites and pathogens including viruses as a stressor that drives pollinator declines. The modified sentences now read: “Pollinator communities worldwide are declining at an alarming rate [1-3]. Emerging viral pathogens have been considered to be major contributors to pollinator losses alongside other drivers such as habitat destruction, increased use of pesticides and fertilizers, climate change and biological factors such as invasive species [1,3,4].” L32-35

  1. Lines 43-44 and line 50: could examples be given about pollinators and associated arthropods?

>>> We have included examples of pollinators and associated arthropods that are infected with bee viruses in this sentence. The modified sentence reads: “Increasing evidence suggest that many of these pathogens are not specific to honey bees and instead are shared between many pollinator species and associated arthropods, including bumble bees and other wild bees [10-12], bee predators such as wasps [10,11], and scavenging insects such as ants, cockroaches and beetles [10,13-15]. L42-45

  1. Line 60: “mite Varroa destructor that vectors bee viruses has not only changed viral disease in honey bees “Could this sentence be explained? You mean that DWV was not disease causing before, but become such due to the mite?

>>> The sentence now reads: “For instance, the global spread of the parasitic mite and bee virus vector, Varroa destructor, affected virus spread and infection levels in honey bees [29], but also caused pathogen shifts in other pollinators and bee predators such as wasps [18,30].” L61-64

Materials and Methods

  1. Line 88: Missing a word?

>>> We added “collected”. This sentence now reads: “Samples of pollinators and associated arthropods (n = 66) were collected using sweep nets or jars when directly collected from abandoned bee hives.” L92-93

  1. “2.5 Phylogenetic analysis”: Please, explain why phylogenetic analysis was done with RdRP for DWV but VP2 for KBV?

>>> Different protein regions were used to detect virus presence (Table S1). We sequenced those amplicons and therefore used sequences covering different fragments of the genome for different viruses in the phylogenetic analyses. To clarify which regions were used in the analysis we added the following sentence: “We used the same genomic regions that were used in the virus detection assay to conduct the phylogenetic analyses, which were a fragment of the RNA dependent RNA polymerase (RdRp) for DWV and the major capsid protein vp3 for KBV.” L156-158

Results

  1. Lines 194-195, 203 and 325: If LhuBLV1 can be found from other arthropods but it cannot be shown to produce mRNA, the virus might still be specific to Argentine ant and be just a contamination in other insects. A virus that does not produce mRNA, does not productively replicate either.

>>> We have corrected this sentence to emphasize that LhuBLV1 detection in arthropods other than Argentine ants could have been caused by contamination. The sentence now reads: “LhuBLV1 was only detected in Argentine ants and associated arthropods in two sites that had ant nests indicating that other species can host this virus, or that its detection on other arthropods could have resulted from contamination. Only Argentine ants tested positive for the mRNA intermediate of LhuBLV1, suggesting that this virus may be L. humile specific.” L209-213

  1. Table 1.: Common names: capital letter or not. Soldier fly?

>>> We thank the reviewer for spotting this inconsistency in capitalisation of common names in the table. All common species names are now in minor case. Table 1, L224

  1. Table2.: Species?

>>> We have corrected “species” to “host” to show that the table header refers to host species. Table 2, L234

  1. “3.4 Viral strain diversity and phylogenetic analysis”: Blast search results should be somewhere to see and “closely matched” should be clarified -what does it mean in numbers? Were complete genome sequences or RdRP/capsid “Blasted” for BQCV, moku and LhuBLV1? (methods?) Was there really only one matching sequence for BQCV?

>>>We have added the percentage identity and query coverage to the BLAST search results and now state which part of the genome was covered by the sequences and used for the BLAST searches. The sentences now read: “BLAST searches showed that BQCV most closely matched the polyprotein of BQCV found in a honey bee in Lithuania (KP223790) (96% identity, 100% query cover). The MKV sequences from 10 different species showed little sequence variation, all sequences derived in this study closely matched (99-100% identity, 97-100% query cover) the MKV polyprotein found in Vespa velutina in Belgium (MF346349) and Vespula pensylvanica from Hawaii (KU645789). LhuBLV1 most closely matched putative RdRp complex gene of the only available LhuBLV1 sequences on GenBank (MH213237) from Argentine ants from Spain (100% identity, 95% query cover).” L248-255

Discussion

  1. Line 301: It is not shown in this manuscript that these viruses cause harm to the host and thus parasite should be removed.

>>> We have changed “parasitizing” to “infecting” in the sentence to emphasise that we only tested for active virus infections not effects on host species. L333

Line 303: It was not shown that LhuBLV1 is replicating because there was not such an assay. You showed that the virus produces mRNA.

>>> We have corrected this sentence and replaced “active replications” with “active infections”. This sentence now reads: “To our knowledge, this work represents the first study to confirm active LhuBLV1 and MKV infections in arthropod hosts by detecting the positive and negative viral strand intermediate, respectively.” L333-335

  1. Line 306. “Or regulatory mechanisms may facilitate viral replication at a different time”. Clarify/elaborate this sentence.

>>> The sentences has been modified to clarify that virus replication may occur in a species but not at the time we tested for virus replication. The sentence now reads: “Virus replication in novel hosts is a key contributor in disease emergence, nevertheless even hosts without active virus replication may show active virus replication at a different time [54] or can be contaminated with infectious viral particles that contribute to disease spread [6].” L335-338

Line 310: although.

>>> The spelling of “although” has been corrected. L342

  1. Line 358: are.

>>> We have removed “are” from this sentence.

Reviewer 2 Report

This study investigated potential transmission of several viruses between honey bees and arthropod species found in association with honeybee colonies, including wasps, cockroaches, spiders and Argentine ants. RT-PCR was used to screen for the presence of the honey bee viruses: Deformed wing virus (DWV), Kashmir bee virus (KBV), Black queen cell virus (BQCV), Moku virus (MKV); as well as an Argentine ant virus Linepithema humile bunya-like virus 1. This study showed that DWV, KBV and MKV were present in other arthropod species, including spiders, cockroaches. Careful and thorough phylogenetic analysis of the viral RT-PCR products sequences from honeybees and other hive associated-arthropod species showed their clustering according to geographic origin rather than according to the insect species from which they were isolated, thereby suggesting “frequent interspecies virus transmission”.

In addition, the authors detected negative-strand RNA (i.e. replicative intermediates of positive strand RNA viruses presence of which indicated replication of viral RNA) of DWV, KBV, MKV in a number of species including spiders and Argentine ants. Based on this, it was concluded that DWV, KBV, MKV replicate in these arthropod species.

The main question to the MS relates to interpretation of the viral negative-strand RNA detection in other species which is the main argument in support of very wide host range of these viruses. The paper should discuss the possibility that the detected negative-strand RNA of DWV, KBV and MKV could be present in the ingested honeybee tissues (intact cells) where replication of these viruses took place. For example, in a recent study by Posada-Florez et al (2019) “Deformed wing virus type A, a major honey bee pathogen, is vectored by the mite Varroa destructor in a non- propagative manner.” Scientific Reports 9:12445, https://doi.org/10.1038/s41598-019-47447-3, it was shown that the loads of negative-strand RNA of DWV type A in the Varroa mites decreased to undetectable levels following several passages on the honey bees with low DWV-A levels. This study also demonstrated that honeybee mRNAs could be detected in the in the Varroa mites by RT-PCR and RNA-seq, which was consistent with the suggestion that intact honeybee cells could be ingested, agreeing with the recent finding that Varroa mites eat honeybee cells (Ramsey, S. D. et al. 2019, Varroa destructor feeds primarily on honey bee fat body tissue and not hemolymph. PNAS  116, 1792–1801, https://doi.org/10.1073/pnas.1818371116 ). Argentine ants are known to eat honeybee brood and even adult bees(e.g. Buys, B. 1990. “Relationships between Argentine ants and honeybees in South Africa”. In: Applied Myrmecology, a World Perspective, pp. 519–524.). Moreover, this study itself showed that “all ants tested positive for DWV were sampled from the nest in an apiary (lines 195-196). 

The manuscript should discuss (e.g. considering recent Posada-Florez et al 2019 paper), that detection of the negative-strand RNA of DWV, KBV, MKV in the hive associated arthropods, which are known to eat honeybees, is necessary but not sufficient to conclude that these arthropod species are hosts to them. Additional experiments in controlled laboratory environment should be carried out including, assessing dynamics of these viruses following infection, detecting the virus negative strand following rearing without access to honeybees.

Replicating viruses might be present in the host at high levels, therefore quantification of the virus loads by qRT-PCR might also provide additional clues. But the study (Material and Methods, section 2.2.)  should include the sensitivity level of the PCR assay for sets of primers used. This the PCR detection threshold for 30 cycles (lines 107-109) would allow to determine minimal copy number of the viral RNA genomes for the individuals tested positive (Table 1, lines 212- 213).

Author Response

Reviewer 2:

This study investigated potential transmission of several viruses between honey bees and arthropod species found in association with honeybee colonies, including wasps, cockroaches, spiders and Argentine ants. RT-PCR was used to screen for the presence of the honey bee viruses: Deformed wing virus (DWV), Kashmir bee virus (KBV), Black queen cell virus (BQCV), Moku virus (MKV); as well as an Argentine ant virus Linepithema humile bunya-like virus 1. This study showed that DWV, KBV and MKV were present in other arthropod species, including spiders, cockroaches. Careful and thorough phylogenetic analysis of the viral RT-PCR products sequences from honeybees and other hive associated-arthropod species showed their clustering according to geographic origin rather than according to the insect species from which they were isolated, thereby suggesting “frequent interspecies virus transmission”.

In addition, the authors detected negative-strand RNA (i.e. replicative intermediates of positive strand RNA viruses presence of which indicated replication of viral RNA) of DWV, KBV, MKV in a number of species including spiders and Argentine ants. Based on this, it was concluded that DWV, KBV, MKV replicate in these arthropod species.

>>> We thank the reviewer for their detailed and thoughtful comments and appreciate the literature suggested.

The main question to the MS relates to interpretation of the viral negative-strand RNA detection in other species which is the main argument in support of very wide host range of these viruses. The paper should discuss the possibility that the detected negative-strand RNA of DWV, KBV and MKV could be present in the ingested honeybee tissues (intact cells) where replication of these viruses took place. For example, in a recent study by Posada-Florez et al (2019) “Deformed wing virus type A, a major honey bee pathogen, is vectored by the mite Varroa destructor in a non- propagative manner.” Scientific Reports 9:12445, https://doi.org/10.1038/s41598-019-47447-3, it was shown that the loads of negative-strand RNA of DWV type A in the Varroa mites decreased to undetectable levels following several passages on the honey bees with low DWV-A levels. This study also demonstrated that honeybee mRNAs could be detected in the in the Varroa mites by RT-PCR and RNA-seq, which was consistent with the suggestion that intact honeybee cells could be ingested, agreeing with the recent finding that Varroa mites eat honeybee cells (Ramsey, S. D. et al. 2019, Varroa destructor feeds primarily on honey bee fat body tissue and not hemolymph. PNAS  116, 1792–1801, https://doi.org/10.1073/pnas.1818371116 ). Argentine ants are known to eat honeybee brood and even adult bees(e.g. Buys, B. 1990. “Relationships between Argentine ants and honeybees in South Africa”. In: Applied Myrmecology, a World Perspective, pp. 519–524.). Moreover, this study itself showed that “all ants tested positive for DWV were sampled from the nest in an apiary (lines 195-196). 

>>> We now discuss the possibility of detecting false positives in the replication assay as a result of recently digested tissue of other species and include all literature suggested by the reviewer.

>>> The following paragraph has been added to the discussion: “We confirmed DWV, KBV and MKV replication in a number of species indicating that these species possibly act as biological vectors that facilitate disease emergence. However, with pollinators and associates collected in the field we cannot exclude the possibility of detecting viruses and negative strand intermediates in the gut contents that come from other infected insects. KBV and MKV replication was confirmed in species not typically known to eat bees such as Polistes wasps and bumble bees. However, some cockroaches, crickets, wasps, and ants that tested positive for virus replication were found in close proximity to bee hives and are species known to scavenge or prey on honey bees [30,33]. Recently consumed viral particles from infected honey bees can cause false positives in replication assays. For instance, Varroa mites which exclusively feed on honey bee tissue [76] and vector DWV are assumed to act as a biological vector that propagates DWV [39,77]. Yet, new research suggest that DWV may not replicate in mites cells, but in honey bee cells recently consumed by mites [78]. Nevertheless, feeding experiments have shown that scavenger species like the ant Myrmica rubra are found positive for the negative strand of DWV for up to 13 weeks after consuming infected honey bees [14], which indicates that DWV actively infects ants. Whether honey bee scavengers are able to spread viruses without becoming infected themselves [79] or aid in reducing virus transmission by removing infectious carcasses [80] remains to be tested. Detection of negative strand intermediates is an essential step in identifying host species but only controlled infection experiments allow to determine virus dynamics in these hosts and the potential for disease emergence.” L406-423

>>> We also highlight and discuss that DWV was only detected in Argentine ants from apiaries and that DWV detection in this species could have been caused by ants scavenging in apiaries. “We only detected DWV in Argentine ants from apiaries, which could indicate that ants acquire the virus from bees, potentially by scavenging in apiaries. L380-383

The manuscript should discuss (e.g. considering recent Posada-Florez et al 2019 paper), that detection of the negative-strand RNA of DWV, KBV, MKV in the hive associated arthropods, which are known to eat honeybees, is necessary but not sufficient to conclude that these arthropod species are hosts to them. Additional experiments in controlled laboratory environment should be carried out including, assessing dynamics of these viruses following infection, detecting the virus negative strand following rearing without access to honeybees.

>>> We thank the reviewer for suggesting this relevant paper. We included the Posada-Florez et al. (2019) in L417-419 and discuss the need for controlled feeding experiments to determine virus dynamics in new host species in L421-423 (see response to comment above).

Replicating viruses might be present in the host at high levels, therefore quantification of the virus loads by qRT-PCR might also provide additional clues. But the study (Material and Methods, section 2.2.)  should include the sensitivity level of the PCR assay for sets of primers used. This the PCR detection threshold for 30 cycles (lines 107-109) would allow to determine minimal copy number of the viral RNA genomes for the individuals tested positive (Table 1, lines 212- 213).

>>> We appreciate the comment and agree that quantitative RT-PCR could provide valuable information on the severity of virus infections in arthropods. However, we believe that quantifying virus infections in novel arthropod hosts should be the work of future studies. In our study we were primarily interested in identifying pollinators and associated that are infected by viruses (L70-71). 

Reviewer 3 Report

Dobelmann et al. report on a straightforward study of arthropod virus prevalence and distribution. The work adds to an important and growing literature on the subject. I have one major criticism about the phylogenetic method and overall, the paper needs copy editing.

Major comment: I would not use BEAST to construct the phylogeny, or I would set BEAST not to use isolation date as information in determining the structure of the tree. The reason is that the main conclusion of the manuscript is that viruses cluster on the phylogeny by region, however, if different sites were sampled at different times, and age of the sample was used to build the phylogeny, then the software will try to cluster samples with similar dates and the authors could find spurious associations between phylogeny and region. This possibility must be addressed in the manuscript. The authors need to re-run this analysis and show that the pattern holds.

Comments by line number

15 – comma after Firstly

17 – This sentence is a fragment, there’s no subject. Specify what was included, e.g. primer sets that amplify these viruses’ RNA/DNA.

67-69 – This is a run-on sentence, and I do not see how these hypotheses are related. I think the authors need to point out in this section that the predominant pattern in the phylogeny should not be host-associations if they can freely move between hosts, but instead, it may show other relationships, such as with space.

69 – comma after firstly like there’s a comma after secondly in line 76

86 – what does ‘more’ refer to?

88 – ‘when directly’ a word is missing between these two words, perhaps insert ‘sampled’

89 – I’m not a bee expert, but I think the term beehive is one word, not ‘bee hive’.

This manuscript has a ton of typos. I stopped commenting on them after line 89.

352 – viruses may also benefit each other and interact mutualistically

Author Response

Reviewer 3:

Dobelmann et al. report on a straightforward study of arthropod virus prevalence and distribution. The work adds to an important and growing literature on the subject. I have one major criticism about the phylogenetic method and overall, the paper needs copy editing.

>>> We thank the reviewer for their thoughtful and relevant comments.

 Major comment: I would not use BEAST to construct the phylogeny, or I would set BEAST not to use isolation date as information in determining the structure of the tree. The reason is that the main conclusion of the manuscript is that viruses cluster on the phylogeny by region, however, if different sites were sampled at different times, and age of the sample was used to build the phylogeny, then the software will try to cluster samples with similar dates and the authors could find spurious associations between phylogeny and region. This possibility must be addressed in the manuscript. The authors need to re-run this analysis and show that the pattern holds.

>>> We appreciate the reviewers comment on the association between sampling date and sampling region. To address this issue we have rerun the phylogenetic analysis. The phylogenetic trees without dated tips are now included in the supplemental materials Figure S2 and Figure S3 and attached with this response letter. We also repeated the BatS analysis which resulted in similar observations with strong clustering observed within geographical regions. These tables are now included in the supplemental materials under Table S3 and Table S4.

>>> For fast evolving RNA viruses dated phylogenetic analyses are commonly used to time calibrate phylogenies, this is now mentioned in the methods: “Tip-dated phylogenies are commonly used to reconstruct RNA viral evolution [50,51], however, there was a chance that sampling date and sampling region in our data set would be connected. Therefore, we repeated the analysis without using the tip-date calibration and compared dated and non-dated phylogenies.” L182-186

>>> Throughout the results we refer to the supplemental Figures S3 and S3 that show the non-dated phylogenies and Tables S3 and S4 with the BatS analysis of the non-dated phylogenies. As there were no major differences observed we feel confident that the tip-dated analysis can be used for our data set.

Comments by line number

15 – comma after Firstly

>>> We have added a comma in L15.

17 – This sentence is a fragment, there’s no subject. Specify what was included, e.g. primer sets that amplify these viruses’ RNA/DNA.

>>> We have reworded this sentence. “We tested for the black queen cell virus (BQCV), Deformed wing virus (DWV), and Kashmir bee virus (KBV) that were initially detected in bees, and the two recently discovered Linepithema humile bunya-like virus 1 (LhuBLV1) and Moku virus (MKV).” L17-19

67-69 – This is a run-on sentence, and I do not see how these hypotheses are related. I think the authors need to point out in this section that the predominant pattern in the phylogeny should not be host-associations if they can freely move between hosts, but instead, it may show other relationships, such as with space.

>>> We have modified this sentence and now state the two hypotheses of our study in separate sentences. It now reads “In this study, we tested the hypothesis that a range of pollinators and associated arthropods, including honey bees and Argentine ants, are infected by the same viral pathogens. Furthermore, we hypothesised that frequent inter-species viral transmission results in viral phylogenies that are predominantly associated with geographic origin instead of being associated with host species.” L69-72

69 – comma after firstly like there’s a comma after secondly in line 76

>>> We have added the comma after firstly in now L72.

86 – what does ‘more’ refer to?

>>> In this sentence “more” refers to the number of honey bee and Argentine ant sampled in comparison to other species sampled. To clarify we have changes this to: “more frequently than other arthropod species” L90

88 – ‘when directly’ a word is missing between these two words, perhaps insert ‘sampled’

>>> We have modified this sentence and added the word “sampled”. The sentence now reads “when directly collected from abandoned bee hives.” L92-93

89 – I’m not a bee expert, but I think the term beehive is one word, not ‘bee hive’.

>>> We have changed “beehive” and “beehives” to “bee hive” and “bee hives”, respectively, throughout the manuscript. L46, L91, L93, L377, L412

 This manuscript has a ton of typos. I stopped commenting on them after line 89.

>>> We have revised and thoroughly proof-read the manuscript.

 352 – viruses may also benefit each other and interact mutualistically

>>> We mention that viruses can have positive effects on other viral infections in L385-386: “Interactions within hosts and, for example, the order in which pathogens are acquired may have large positive or negative effect on a secondary infections [68,69].”

Round 2

Reviewer 3 Report

Authors have addressed my concerns